# The Lysyl Oxidase G473A Polymorphism Exacerbates Oral Cancer Development in Humans and Mice

**DOI:** 10.3390/ijms24119407

**Published:** 2023-05-28

**Authors:** Yaser Peymanfar, Faranak Mahjour, Neha Shrestha, Ana de la Cueva, Ying Chen, Shengyuan Huang, Kathrin H. Kirsch, Xiaozhe Han, Philip C. Trackman

**Affiliations:** 1The Forsyth Institute, 245 First Street, Cambridge, MA 02142, USA; ypeymanfar@forsyth.org (Y.P.); ychen@forsyth.org (Y.C.); 2Department of Translational Dental Medicine, Henry M. Goldman School of Dental Medicine, Boston University, 700 Albany Street, Boston, MA 02118, USA; faranakmahjour@gmail.com (F.M.); neha014@gmail.com (N.S.); 3Department of Biochemistry, School of Medicine, Boston University, 72 East Concord Street, Boston, MA 02118, USA; anadelacueva.her@gmail.com (A.d.l.C.); kkirsch@mgh.harvard.edu (K.H.K.); 4Department of Oral Science and Translational Research, College of Dental Medicine, Nova Southeastern University, Fort Lauderdale, FL 33314, USA; shuang@nova.edu (S.H.); xhan@nova.edu (X.H.)

**Keywords:** oral cancer, polymorphism, lysyl oxidase, human, mouse model, 4 NQO, extracellular matrix, tumor suppressor, pathogenesis

## Abstract

Oral cancer is primarily squamous-cell carcinoma with a 5-year survival rate of approximately 50%. Lysyl oxidase (LOX) participates in collagen and elastin maturation. The propeptide of LOX is released as an 18 kDa protein (LOX-PP) in the extracellular environment by procollagen C-proteinases and has tumor-inhibitory properties. A polymorphism in the propeptide region of *LOX* (rs1800449, G473A) results in a single amino acid substitution of Gln for Arg. Here we investigated the frequency of rs1800449 in OSCC employing TCGA database resources and determined the kinetics and severity of precancerous oral lesion development in wildtype and corresponding knockin mice after exposure to 4-nitroquinoline oxide (4 NQO) in drinking water. Data show that the OSCC is more common in humans carrying the variant compared to the wildtype. Knockin mice are more susceptible to lesion development. The immunohistochemistry of LOX in mouse tissues and in vitro studies point to a negative feedback pathway of wildtype LOX-PP on LOX expression that is deficient in knockin mice. Data further demonstrate modulations of T cell phenotype in knockin mice toward a more tumor-permissive condition. Data provide initial evidence for rs1800449 as an oral cancer susceptibility biomarker and point to opportunities to better understand the functional mechanism of LOX-PP cancer inhibitory activity.

## 1. Introduction

Lysyl oxidase (LOX) catalyzes the final enzyme reaction required for the subsequent crosslinking of elastin and collagen to promote extracellular matrix maturation [1]. LOX is synthesized and secreted as a 50 kDa proenzyme (Pro-LOX), which is cleaved extracellularly to a 30 kDa active LOX enzyme and an 18 kDa lysyl oxidase propeptide (LOX-PP) by procollagen C-proteinases [2]. Interestingly, LOX enzyme activity promotes tumor invasiveness and metastasis by modulating the extracellular matrix surrounding the tumor and actively simulating the formation of metastatic niches [3], while LOX-PP has tumor-growth inhibitory properties [4,5]. The highly disordered LOX-PP protein has multiple binding partners and several mechanisms of action [6]. LOX-PP inhibits several signaling pathways, including RAS, FGF-2, and FAK signaling. Additionally, LOX-PP targets DNA-repair pathways [4,5,7]. LOX-PP inhibits breast, pancreatic, prostate, and lung cancer, and hepatocellular carcinoma cell growth [5,8,9,10]. Furthermore, LOX-PP overexpression inhibits the formation of breast and prostate xenograft tumors [7,11]. By contrast, increased LOX enzyme activity derived from LOX or other *LOX* gene family members, such as LOXL2, promotes cancer and is associated with poor clinical outcomes. 

A single-nucleotide polymorphism (SNP) in the *LOX* gene G473A results in an arginine (Arg) 158 substitution by glutamine (Gln) in a highly conserved region of the LOX-PP sequence. The 473A allele of the *LOX* gene in European, Asian, Sub-Saharan African, and African American populations in the HapMap database was previously found at an average frequency of 24.6%. This naturally occurring G473A polymorphism (rs1800449) appears to attenuate the ability of LOX-PP to function as a tumor suppressor [12]. The SNP was reported to associate with higher breast-cancer risk in European women [13] and African American women carrying this allele have a higher risk for triple-negative breast cancer [12]. It also correlates with breast cancer and ovarian cancer in the Chinese Han population [14,15]. In another study, the G473A polymorphism was associated with a higher risk of lung and colon cancer in cigarette smokers in a North Chinese population [16]. These data show an important role of the *LOX* G473A polymorphism in the development of solid cancers. However, the effect of this polymorphism on the occurrence of oral cancer remains unknown.

Oral squamous-cell carcinoma (OSCC) represents over 90% of oral cavity cancers [17,18]. Despite modern treatments including surgery, radiotherapy, and chemotherapy, and treatment with novel biologics, the five-year survival rate is approximately 50%. The poor prognosis of OSCC is mostly due to the late diagnosis of the disease after it reaches advanced stages [19]. In addition to gene mutations, OSCC has strong associations with environmental factors such as alcohol and tobacco abuse and human papillomavirus infections (HPV16-18) [17,18]. The genetic susceptibility of OSCC is still largely undefined because of the multifactorial nature of OSCC. The present study determined whether the G473A polymorphism results in increased oral lesion development in a chemically induced oral-cancer mouse model. In this study, we analyzed the association of the single nucleotide polymorphism at G473A (rs1800449) within the LOX propeptide in the incidence of human tongue OSCC using available TCGA data. For independent confirmation, we employed the homozygous knockin mouse that corresponds to the polymorphic human Arg/Gln polymorphism at residue 158 [20], which corresponds to residue 152 in mice. Here, we investigated the susceptibility of these mice to the DNA adduct-forming agent, 4-nitroquinoline-1 oxide (4 NQO), which is a major mutagen in tobacco smoke, according to established protocols for studying oral cancer [21]. The 4 NQO mouse protocol recapitulates human oral squamous-cell cancer development from the normal tongue epithelium and progresses through dysplastic lesions that ultimately develop into OSCC and is well suited to assess for susceptibility differences between wildtype and genetically modified mice such as our Arg to Gln LOX-PP knockin mice compared to wildtype.

Our analyses of the TCGA database indicate a significant association of OSCC with rs1800449 human polymorphism. Supporting this, we found that knockin mice are much more susceptible to oral lesion development with poor outcomes compared to wildtype mice. LOX was strongly upregulated in knockin 4 NQO-treated mouse lesions compared to wildtype, suggesting that LOX expression is downregulated by Arg LOX-PP but not Gln LOX-PP. Our in vitro studies further support that wildtype LOX-PP (Arg LOX-PP) inhibits LOX expression in an oral-tumor cell line, while mutant LOX-PP (Gln LOX-PP) has a reduced inhibitory effect. Our findings point to a novel unexpected feedback mechanism for wildtype LOX-PP suppression of LOX production that is disrupted by the variant. Since active LOX enzyme levels are strongly associated with poor clinical outcomes, our data provide at least one possible pathway for the observed poor cancer outcomes associated with rs1800449 polymorphism. Preliminary evidence for differential heightened immune suppression in knockin mice is also presented. 

## 2. Results

### 2.1. Increased Human Oral-Cancer Incidence in rs1800449 Polymorphic-Variant Subjects

Analyses of the genomic-sequence data sets associated with non-specified oral sites and tongue squamous-cell neoplasms (TCGA-HNSC) retrieved from the GDC port revealed that the frequency of the LOX-PP rs1800449 polymorphism in oral-cancer patients (26.92%) is significantly higher compared to the frequency of rs1800449 in both the American population (17.57%) *p* = 0.0288 and the global (16.98%) *p* = 0.0117 population data incidence of oral cancer in patients, according to the NIH HapMap (a haplotype map) of the human genome data set (Figure 1). The data suggest that there is a strong association between LOX-PP polymorphism and the development of oral cancer. 

### 2.2. Increased Lesion Occurrence in Arg to Gln LOX-PP Knockin Mice

Four-month-old LOX-PP knockin mice (Pro-LOX Gln152/Gln152), whose sequence mimics human rs1800449 G473A polymorphism, and wildtype control mice (Pro-LOX Arg152/Arg152) [20] (Figure 2) were randomly given either vehicle or 4 NQO in their drinking water for 16 weeks. All animal cages were reverted to 4 NQO-free drinking water and the mice were monitored for an additional week (Figure 3). 

A total of thirteen Arg to Gln LOX-PP knockin mice and seven wildtype mice were employed. Eight of the knockin mice were treated with 4 NQO and five of them were treated with vehicle. All of the wildtype mice were treated with 4 NQO. It was observed that in the Gln LOX-PP knockin group, the number of mice that developed tongue lesions was significantly higher following the 4 NQO treatment than in the wildtype group treated with 4 NQO at 17 weeks (Table 1). The mice in the LOX-PP knockin group treated with 4 NQO developed lesions with a higher incidence and faster development compared to WT mice treated with 4 NQO (Figure 4). One mouse in the LOX-PP knockin group treated with NQO developed a lesion on the floor of the mouth in the 12th week after treatment with 4 NQO and died two weeks later. Lesions developed in two LOX-PP knockin mice after the 16th week and three LOX-PP knockin mice after the 17th week of initiation of 4 NQO treatment developed cancer lesions on their tongues. Only one mouse in the wildtype group treated with 4 NQO developed tongue lesions by the 17th week. In total, six out of eight mice (75%) in the LOX-PP knockin group treated with 4 NQO developed oral cancer, while only one mouse out of seven (14%) in the 4 NQO-treated wildtype group developed tongue tumors. A chi-square test (3X2 analysis) indicated a significant number of mice with lesions in the Gln LOX-PP knockin group treated with 4 NQO compared to other experimental groups (Table 1) (*p* = 0.008). Fisher’s exact test (2X2 analysis) between knockin mice treated with 4 NQO vs. wildtype (Arg LOX-PP) treated with 4 NQO indicated that the incidence of cancer in the Gln LOX-PP knockin mice treated with 4 NQO is more significant compared to the mice treated with 4 NQO (*p* < 0.05). Fisher’s exact test (2X2 analysis) illustrated a remarkably high incidence of lesions in the Gln LOX-PP knockin mice treated with 4 NQO vs. Gln LOX-PP knockin mice treated with water (*p* < 0.05). At the final time point of this experiment, there was no significant statistical difference for the oral lesion incidence between knockin mice given plain water only and wildtype mice treated with 4 NQO (*p* > 0.05). In the latter group, only one mouse developed a visible tongue lesion. The mice in the knockin group exhibited a greater increase in tongue volumes compared to wildtype after 4 NQO treatment (*p* < 0.05) (Figure 4). These results suggest that LOX-PP polymorphism reduced the ability of LOX-PP to suppress tumor formation in the presence of 4 NQO. 

### 2.3. Histopathology of the Tongue Lesions

Histology sections stained with hematoxylin and eosin of the tongue lesions that developed in both knockin and wildtype mice treated with 4 NQO exhibited similar features of excess squamous cell accumulation, many of which exhibited a loss of close cell-to-cell contacts (red arrows in Figure 5). Specifically, atypical papillary exophytic squamous lesions were observed, some with focal dysplasia. Some cells appeared to be hyperchromatic and there was cellular discohesion with cells exhibiting increased nuclear-to-cytoplasmic ratios. Epithelial hyperplasia was also observed adjacent to the papillary outgrowths. Basal-cell hyperplasia and dysmaturation were observed. This abnormal phenotype was more apparent in the knockin mice (Figure 5). The overall finding from histopathology is that the knockin mice develop lesions characterized by an atypical pappilary exophytic squamous proliferation and general epithelial hyperplasia. Analyses of epithelial thickness showed clear generalized acanthosis in 4 NQO-treated mice with the greatest thickness observed in the 4 NQO-treated knockin mice (Figure 4B and Figure 5). These findings are consistent with dysplastic lesions developing in 4 NQO, both wildtype and knockin mice, with knockin mice showing a stronger phenotype, while untreated knockin mice exhibited a normal morphology. Only one wildtype mouse treated with 4 NQO developed a papillary exophytic squamous proliferation, in sharp contrast to the 4 NQO-treated knockin mice (Table 1 and Figure 5).

Proliferating cell nuclear antigen (PCNA) is a marker of cell proliferation. Immunohistochemistry for PCNA in vehicle-treated knockin mice exhibited normal strong nuclear expression in basal epithelial cells (Figure 6, top panels). By contrast, in 4 NQO treated knockin mice, strong nuclear staining was observed in fibroblasts in the connective tissue layer, consistent with dysplasia (Figure 6, middle panels, red arrows). Interestingly, in wildtype mice treated with 4 NQO, PCNA staining was not observed in fibroblasts, but rather in the refractile suprabasal epithelial cells in the papillary lesion, and adjacent to the papillary lesion in basal and suprabasal cells, suggesting elevated epithelial cell proliferation without as large a development of a stromal dysplastic reaction (Figure 6, bottom panels, blue arrows). 

Lysyl oxidase is secreted as a pro-protein, and expression is typically found both inside cells and cell associated. In samples from vehicle-treated knockin mice, a typical normal staining pattern of expression associated with the basal epithelium was observed, consistent with its function in maintaining the integrity of the basal lamina (Figure 7, top panels, back arrows). Spinous epithelial staining was also observed. In 4 NQO-treated knockin mice staining for lysyl oxidase revealed a high degree of staining in fibroblasts in the stromal compartment and an apparent disorganized arrangement of cells (Figure 7, middle panels, red arrows), consistent with a strong dysplastic phenotype and profibrotic activities of LOX. In addition, a high degree of epithelial cell staining was observed primarily in suprabasal cells and spinous epithelial regions. Interestingly, in 4 NQO treated wildtype mice, epithelial cell staining was weaker and primarily associated with cells that appeared to have intact cell-cell interactions, and not the more fractile epithelial cells in the papillary lesions (Figure 7, middle panel, blue arrows). Wildtype mice treated with 4 NQO had weak staining for lysyl oxidase in and around basal epithelial cells and the connective tissue interface, unlike knockin mice (Figure 7, bottom panels, white arrows). This finding further supports the highly dysplastic nature of the knockin lesions compared to wildtype. In wildtype mice, staining was mainly observed in the suprabasal epithelial regions and little in the connective tissue. (Figure 7, bottom panels, white arrows). 

The findings in Figure 7 suggested that wildtype Arg LOX-PP could inhibit the expression of LOX via negative feedback that is attenuated in Arg > Gln substitution in the pro-peptide domain. To investigate this possibility, we treated cultures of the mouse oral-cancer cell line LY2 with recombinant wild with vehicle type rat Arg LOX-PP or Gln LOX-PP overnight in serum-free medium followed by a Western blot of three cultures each. Data in Figure 8 shows that 50 kDa pro-LOX bands were diminished in Arg LOX-PP-treated cells, while this response was weaker in Gln LOX-PP treated cells.

To begin to assess immune cell differences in knockin vs. wildtype mice, we assessed selected immune-marker expression patterns at the borders of lesions in histologic sections. Specifically, we compared T cell PD-L1/PD-1 expressions in cancer lesions between wildtype mice and knockin mice and CD4^+/−^ and CD8^+/−^ T cell expression. Immunofluorescent staining of 4 NQO tumor lesions indicated that a significant increase in PD-L1 production was observed in the lesions of knockin Gln LOX-PP mice compared to wildtype Arg LOX-PP mice, and this production was not by CD4^+^ T cells. Cancer-cell-associated PD-1-expressing CD4^+^ T cells were relatively high in knockin Gln LOX-PP mice, whereas PD-1-expressing CD8^+^ T cells were decreased in knockin Gln LOX-PP mice compared to wildtype Arg LOX-PP mice. However, PD-L1-expressing CD8^+^ T cells were relatively high in wildtype Arg LOX-PP mice (Figure 9). These results suggest that wildtype Arg LOX-PP, compared to Gln LOX-PP, could differentially modulate immune T cell PD-L1 and PD-1 expression in OSCC toward a more tumor-suppressive environment. 

Taken together, the data presented here from the mouse study indicate that LOX-PP knockin mice are more susceptible to oral cancer development than wildtype mice. In vivo and in vitro data in mice and mouse cancer cells suggest a feedback pathway in which Gln LOX-PP has lost the ability to inhibit LOX expression, while wildtype Arg LOX-PP maintains a more normal LOX expression in a negative feedback pathway. Moreover, differential immune modulation by Arg LOX-PP vs. variant Gln LOX-PP is further implied by our findings. Combined with increased OSCC associated with the LOX polymorphism in humans, data point to an important role for LOX-PP in oral-tissue homeostasis with relevance to the incidence of oral cancer that must be further explored. 

## 3. Discussion

OSCC is one of the most prevalent cancers in the world with poor clinical outcomes and a high recurrence rate [22,23]. Despite modern treatment methods such as chemotherapy, radiotherapy, surgery, and biologics, the overall 5-year survival rate is only 50%, which is mostly due to the late diagnosis of the disease [19]. Several risk factors, including tobacco and alcohol consumption, promote the development of oral cancer [24]; however, the role of genetic factors, including polymorphisms of genes important to OSCC development and progression, is unclear. The LOX family proteins are known as extracellular enzymes that are critical in the biosynthetic maturation of collagen and elastin [25]. However, LOXs were found to be highly expressed in a variety of cancers, resulting in the formation of aggressive tumors, reduced survival rate, and increased metastasis in breast, prostate, colon, and lung cancers [3,26,27,28]. By contrast, the propeptide domain of LOX (LOX-PP) derived from the pro-LOX has been shown to inhibit breast, pancreatic, lung, and prostate cancers [5,9,12,20]. LOX-PP polymorphism G473A (rs1800449) results in a change in amino acid residue 158 from Arg to Gln in humans and was shown to result in enhanced breast cancer development [20]. Here, we analyzed for the first time genomic oral-cancer data from the TCGA database, which mainly represents the U.S data [29,30], and found that the frequency of G > A of G473A is indeed significantly higher in OSCC patients (26.92%) compared to G > A allele frequency (17.57%) in American populations, according to the HapMap human genome project dataset (*p* = 0.0288). The frequency of rs1800449 polymorphism was even more significant (*p* = 0.0117) when genomic cancer data were compared with the global population (16.98%). These data strongly suggest that LOX-PP G473A polymorphism is associated with an increased incidence of oral cancer in patients. 

To examine the role of LOX-PP polymorphism in the incidence and progress of oral cancer, Gln LOX-PP knockin mice (R152Q), which mimic human G473A and have the same Arg > Gln substitution in the propeptide domain, were employed. Oral cancer was chemically induced with 4 NQO in LOX-PP knockin and WT mice. Carcinogen 4 NQO induction of oral cancer in mice has been shown previously as a useful animal model for the study of oral cancer in both male and female mice [31]. The incidence of oral lesions in tongues was observed at a much higher rate (*p* < 0.05) in the Gln LOX-PP knockin group compared to WT mice with the same treatment. No statistically significant differences for lesion detection were observed between knockin mice treated with plain water and WT mice given 4 NQO (*p* > 0.05) under the conditions of the time of treatment in this experiment, as expected. Taken together, these results suggest an important role of the LOX-PP polymorphism in the biogenesis of oral lesions in the presence of the carcinogen 4 NQO. Consistent with our previous data, and data by others, the LOX-PP polymorphism was here demonstrated as a potential biomarker associated for OSCC. 

Lesions developed under the conditions of the current study were determined by a pathologist to exhibit severe dysplasia in the 4 NQO-treated knockin mice, with milder dysplasia developing in the 4 NQO-treated wildtype mice. The 4 NQO model progresses first to hyperplasia, dysplasia, and, finally, to full-blown OSCC, and accurately mimics the development of human oral cancer [21]. For full OSCC development, wild mice must be maintained for 25 weeks [21], while our experimental design here sought to observe lesion development at an earlier time point to assess differences in the relatively early development of these oral lesions. Data suggest an accelerated developmental program that promotes early dysplasia development, and likely the early development of OSCC. 

Immune modulations in the context of LOX and cancer so far have been primarily documenting elevated expression levels of increased LOX enzyme activity and its relationships to increased tissue stiffness due to increased collagen cross-linking and fibrosis [32,33,34]. Immune-cell phenotype has been implicated in these studies as a driving force in regulating cancer development and possibly LOX expression. We are, therefore, developing hypotheses regarding mechanisms of differential modulation of immune-cell phenotypes by wildtype and variant rLOX-PP on immune-cell phenotypes. Here, we wished to begin to assess in vivo whether such modulation could occur. We have provided initial findings that point to a link between the *LOX* polymorphic variant in the enzymatically inactive propeptide region and increased cancer susceptibility and worse outcomes. At least two mechanisms can be envisioned. One, as we have discussed above, is the loss of a feedback inhibition pathway on LOX itself. Another is that LOX-PP or secreted pro-LOX that contains the LOX-PP sequence secreted from tumor cells and associated stromal cells could have unexplored direct interactions and regulatory functions with immune cells in the microenvironment implicated by our data taken together in Figure 7, Figure 8 and Figure 9. These possibilities are under investigation in our laboratories and are focused on the effects of LOX-PPs on cells in the microenvironment. 

Previous studies revealed that LOX-PP, but not LOX, inhibits the Ras-transformed NIH3T3 fibroblast cell line [4]. It was shown that LOX-PP attenuates Her-2/neu-driven tumor development in a nude-mice breast-cancer model, via suppressing the ERK1/2, AKT, and NF-kB signaling pathways [35]. Adenoviral vector-induced LOX-PP expression reduced cell migration and suppressed angiogenic factors MMP2 and MMP9 [10]. Identification of LOX-PP polymorphism with reduced antitumor activity in several cancers may open the venue to understand the role of Arg158 in the antitumor activity of LOX-PP which is diminished in the mutant Gln158 variant. Of note, it was shown that MMP2 can cleave pro-LOX immediately adjacent to Arg158 in the most predominant form of pro-LOX, but this cleavage occurs before Gln158 in a less common form of pro-LOX [36,37,38]. One possibility is that Gln158 could be resistant to MMP2 proteolytic cleavage at residue 158 which then may alter the activity or substrate specificity of pro-LOX in favor of more BMP1/procollagen C-proteinase cleavage at residue 158, resulting in the formation of active LOX, which is well-known to be upregulated in a variety of cancers [39,40]. Therefore, it is possible that LOX and Arg LOX-PP play opposing effects in tumor development and metastasis by this or another still unexplored mechanism. Consistently, our data presented here indicate that overexpression of LOX occurs in connective-tissue cells adjacent to the basement membrane in Gln LOX-PP knockin mice but not in control groups (Figure 7), promoting dysplasia in the knockin mice by failing to block inhibitory pathways in fibroblasts. These findings strongly suggest that wildtype Arg LOX-PP inhibits the expression of LOX which is attenuated in the Gln LOX-PP variant. By conducting the LY2 cell-culture model of oral cancer and treating cells with Arg LOX-PP or Gln LOX-PP recombinant protein, the more pro-LOX protein was observed to be expressed following Gln LOX-PP treatment, while less was expressed after wildtype Arg LOX-PP treatment (Figure 8). This observation is fully consistent with our in vivo findings.

In summary, our in silico and in vivo data indicate that the LOX-PP polymorphism results in increased incidence and severity of oral cancer development. The mouse and cell-culture data strongly suggest a loss of naturally occurring inhibitory activity of the Gln LOX-PP variant compared to wildtype Arg LOX-PP which ultimately results in the elevation of LOX production and tumor development (Figure 10). Our findings indicate that LOX-PP G473A polymorphism could be an important biological marker with a strong association with the oral-cancer phenotype and may open new avenues for future studies and therapeutic approaches. 

## 4. Materials and Methods

### 4.1. LOX G473A Polymorphism in Oral-Cancer Patients

The NCI Genomic Data Commons (GDC) portal was used to access The Cancer Genome Atlas (TCGA) program. Genomic sequences from 157 male and female patients diagnosed with tongue squamous-cell neoplasms and unspecified parts of the tongue (TCGA-HNSC project) were downloaded from the TCGA database, specifically dbGap Study Accession phs000178 [29]. The genome sequences (.BAM files) were selected and sliced to include the region of interest of the lysyl oxidase gene on chromosome 5 between 122,070,800-122,078,200 bp locus, release (GRCh38.p13). To view the SNP sequences, index files (.BAI files) were constructed for all associated BAM files using an open-source software gene pattern (Broad Institute, MIT, Cambridge, MA, USA and University of California, USA). The rs1800449 (LOX-PP R158Q polymorphism) sequences [41] were analyzed for the point mutation (sequence C > T and reference sequence G > A) at 122.077.513 bp site which is located at a highly conserved region of LOX-PP (Figure 2) by employing the open-source Integrative Genomics Viewer (IGV) software, version 2.12.3 (Broad Institute, MIT, Cambridge, MA, USA and University of California, USA). TCGA datasets are primarily collected from U.S. institutions [29,30]. Analyzed data were, therefore, compared to allele frequency of rs1800449 data from the NIH HapMap (a haplotype map) of the human genome data set from the American population. TCGA is recognized as the only comprehensive cancer genomic database. Therefore, in a separate analysis, TCGA data was also compared to the global frequency of rs1800449 according to NIH HapMap [41]. Chi-square statistical analysis was performed, and the *p* values were determined using GraphPad Prism 9.5.1, Clarivate, Philadelphia, PA, USA. 

### 4.2. Animals Study and 4 NQO Treatment

Animal protocols were reviewed and approved by the Boston University Medical Center Institutional Animal Care and Use Committee (IACUC). Homozygous LOX knockin mice in the C57BL/6 background were generated in the *LOX* gene (Arg152Gln change), as we described previously [20]. 

Thirteen knockin mice and seven wildtype mice beginning at 4 months old were used in this study. Male and female mice were analyzed together. The 4-Nitroquinoline-1-oxide (4 NQO) was obtained from Sigma-Aldrich (catalog # N8141) and was dissolved in propylene glycol (W294004, Sigma-Aldrich) at a concentration of 4 mg/mL and stored at 4 °C. Then, the solution of 4 NQO was added to red opaque sterilized water bottles for mice containing drinking water to obtain a final concentration of 100 μg/mL. This concentration of 4 NQO restricts lesion development to the tongue and oral cavity in both male and female mice that experience primary contact with the mutagen [31]. The water was changed once a week during the 16 weeks of the 4 NQO treatment. Regular chow and soft diet gel containing nutrients (Diet Gel76A, ClearH20, Westbrook, ME, USA) were provided for the mice throughout the experimental period. Mice were monitored for food consumption and weight loss.

Animals were euthanized by isoflurane inhalation, followed by cervical dislocation on the 17th week after initiation of treatment with NQO. The oral cavity, esophagus, and stomach were assessed for any pathological lesions or tumors. Tongues were fixed in 4% paraformaldehyde, embedded in paraffin, and sectioned into 5-μm sections for histology and immunohistochemistry.

### 4.3. Histology and Immunohistochemistry

Sections were deparaffinized, rehydrated, and stained with H&E for histopathology. Sections were stained overnight with the following primary antibodies; PCNA (dilution, 1:500, #ab92552, Abcam, Wlatham, MA, USA), and LOX antibody at 1 µg/mL [42]. The slides were incubated with a secondary biotinylated goat antirabbit immunoglobulin antibody, as recommended by the manufacturer (Vector Laboratories, Newark, CA, USA). Control slides were generated by incubating the sections with rabbit IgG (Vector Laboratories) instead of primary antibody and showed no staining at the same IgG concentrations. The slides were developed with 3, 3-diaminobenzidine (DAB Peroxidase substrate kit, Vector Laboratories, Inc.). Images were captured on a digital slide scanner and processed using CaseViewer 2.3 software (3D Histotech, Budapest, Hungary). 

The epithelium thickness of the tongues was measured In 4 mice per group (10 measurements per mouse). The epithelium thickness was measured at intervals from the top to the base of the epithelium layer and from the top of the tongue towards the middle every 0.5 mm, overall, 4.5 mm in length. Data for each individual mouse were then averaged to produce the final thickness data point for each mouse. 

Sections were subjected to immunofluorescence staining to detect the phenotype of T cells and the expression of PD-1 and PD-L1. FITC anti-mouse CD4 (dilution, 1:500, catalogue #100406, Biolegend, San Diego, CA, USA)), Alexa Fluor 488 antimouse CD8 (dilution, 1:500, catalogue #126628, Biolegend, San Diego, CA, USA), antimouse PD-1 (dilution, 1:500, #135202, Biolegend, San Diego, CA, USA), and antimouse PD-L1 (dilution, 1:500, #124302, Biolegend, San Diego, CA, USA) antibodies were incubated with tissue sections overnight at 4 °C, then incubated with Alexa Fluor 647 anti-rat antibody (dilution, 1:500, #407512, Biolegend, San Diego, CA, USA) for 1 h. Sections were counterstained with DAPI for cell-nucleus staining. Images were captured and recorded by EVOS M5000 Imaging System (Invitrogen, Carlsbad, CA, USA). Subsequent image analysis was performed using ImageJ software (Version, 1.53t). Briefly, the double-positive areas of Alexa Fluor 647 and FITC were selected and counted in each region of interest (ROI), which is the subepithelial connective tissue area adjacent to the squamous cancerous lesion. After measuring the area of the ROI, the number of positive cells was presented as cells/mm^2^, according to the defined ROI.

### 4.4. Cell Culture and Western Blots

To examine the effects of mutant and wildtype LOX-PP on LOX expression, mouse oral-cancer cell line, LY2 cells [43] were cultured and, at confluency, cells were treated overnight in serum-free medium with 8 µg/mL recombinant wildtype rat Arg LOX-PP or Gln-LOX-PP. Cell layers were extracted into a 5X SDSPAGE sample buffer. Equal amounts of proteins were applied to 10% SDS PAGE gels and transferred to PVDF membranes at 66 mA overnight. Membranes were washed in TBS-T (20 mM Tris, 150 mM NaCl, and 0.01% Tween-20) and then blocked for an hour in 5% nonfat dry milk in TBS-T and incubated with primary antibody overnight at 4 °C. Antibodies employed were rabbit anti-LOX (NSJ BioReagents #32045, San Diego, CA, USA), or rabbit mAb β-actin (Cell Signaling, catalogue #4970, Danvers, MA USA). Membranes were washed in TBST 3 times for 10 min and were incubated with antirabbit IgG, HRP-linked antibody (#7074, Cell Signaling, San Diego, CA, USA) for 2 **h** at room temperature. HyGlo Quick spray (Denville Scientific, catalogue #1001354, Holliston, MA, USA) or SuperSignal™ West Femto Maximum Sensitivity Substrate (#34095, Thermo Fisher Scientific, Waltham, MA, USA) were used to visualize the signals using a chemiluminescence detector (G:BOX, Syngene, Bangalore, India). Using ImageJ software, version 1.53t, the band intensity of proteins was analyzed from three independent samples/group and normalized to their respective β-actin signals after stripping and reprobing. 

### 4.5. Statistical Analysis

The chi-square test, Fisher’s exact test, one-way ANOVA, two-way ANOVA, and post hoc tests were performed using Graphpad Prism (version 9.5.1 for windows, GraphPad Software, San Diego, CA, USA). Significance was considered when *p* < 0.05. In the graphs, asterisks on bars are representative of significant differences between the indicated columns compared to the control. 

## Figures and Tables

**Figure 1 ijms-24-09407-f001:**
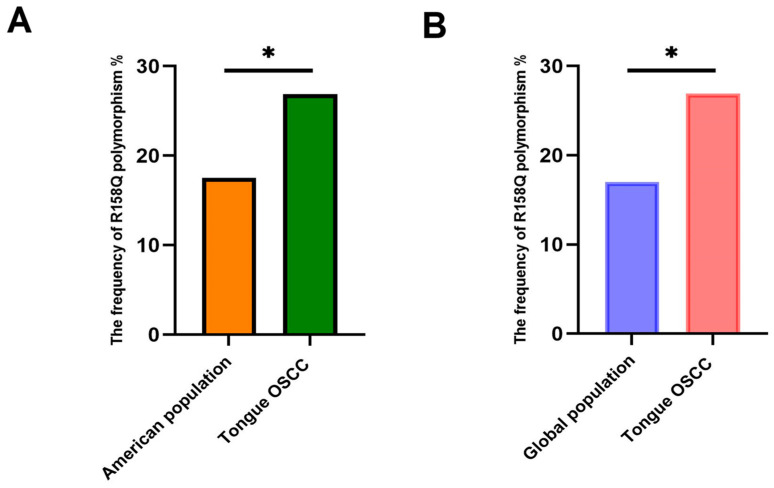
High frequency of rs1800449 (LOX-PP R158Q polymorphism) in OSCC. Chi-square tests: (**A**) *p* = 0.0288; *n* = 156 for cancer patients, and *n* = 770 for G > A allele frequency (NIH HapMap haplotype map) of the American population. (**B**) *p* = 0.0117; *n* = 156 for cancer patients, and *n* = 1890 for G > A allele frequency (NIH HapMap haplotype map) of the data set from the human global population. * *p* < 0.05.

**Figure 2 ijms-24-09407-f002:**
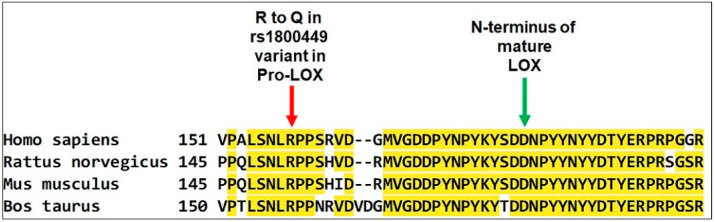
Conserved sequence around human residue R158 in pro-LOX, which when mutated to R158Q compromises LOX-PP tumor-suppressor function. Only the partial C-terminal end of the 14 kDa LOX-PP sequence plus the first few residues of mature LOX domains are shown. The procollagen-C proteinase processing site of pro-LOX that separates LOX-PP from active mature LOX is shown in green. Yellow highlights indicate conserved amino acid sequences.

**Figure 3 ijms-24-09407-f003:**
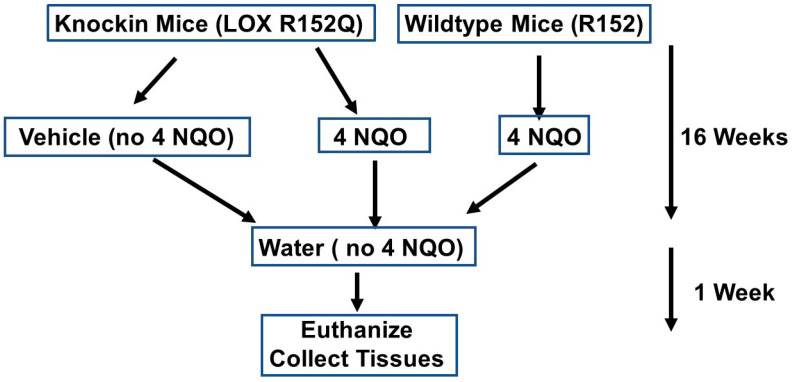
Experimental design.

**Figure 4 ijms-24-09407-f004:**
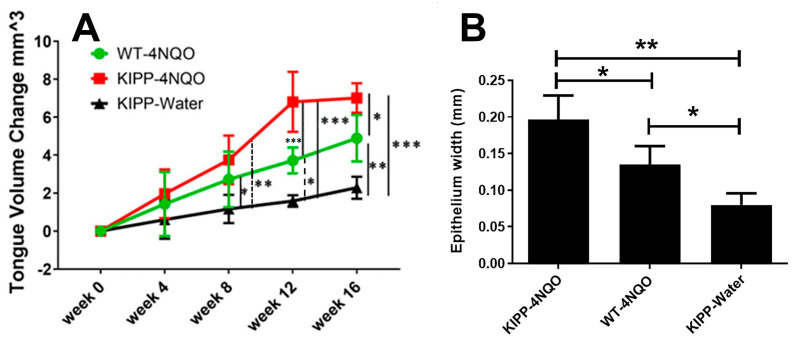
Tongue volume measurements in experimental mice. (**A**) tongue volumes; (**B**) epithelial thickness. Wildtype mice (WT); knockin mice (KIPP). The error bars indicate +/− SD. Two-way ANOVA, *p* < 0.05, Tukey’s multiple comparison test, * *p* < 0.05, ** *p* < 0.001, *** *p* < 0.0001 indicate that the mice in the knockin group developed larger volumes of tongues compared to wildtype after 4 NQO treatment.

**Figure 5 ijms-24-09407-f005:**
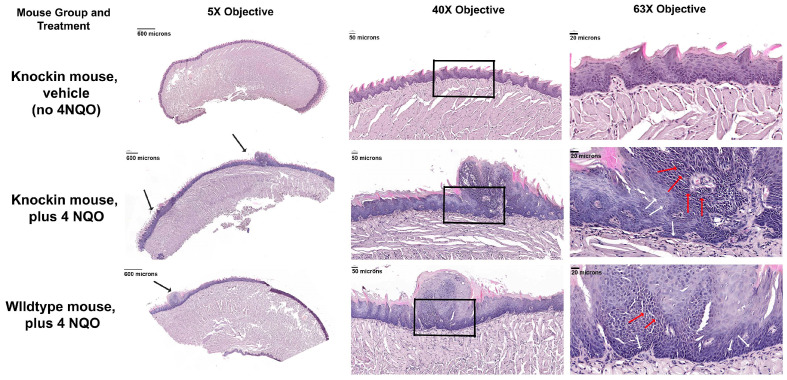
Hematoxylin and eosin staining of mouse tongues show dysplastic lesions generated in 4 NQO-treated especially in knockin mice with an attenuated response in the one wildtype mouse that developed a papillary lesion. Black arrows show lesion formation. Red arrows show hyperchromatic discohesive cells with increased nuclear to cytoplasmic ratios and white arrows show cohesive epithelial cells. Boxes outline areas shown at higher magnification. Images are representative of 4 mice per group subjected to histology and immunohistochemistry.

**Figure 6 ijms-24-09407-f006:**
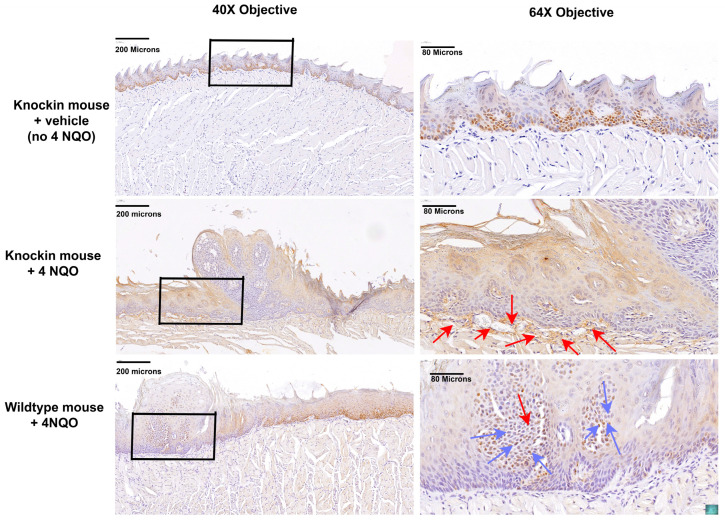
Differential PCNA expressions in mouse groups. Knockin mice treated with vehicle exhibit normal basal epithelial expression of PCNA (top panels), while 4 NQO treated mice exhibit lesions with connective-tissue stromal cell expression (middle panels, red arrows), while wildtype mice treated with 4 NQO exhibit suprabasal and basal epithelial staining (bottom panels, blue arrows). The boxes identify regions shown at higher magnification in the right columns. Images are representative of 4 mice per group subjected to histology and immunohistochemistry.

**Figure 7 ijms-24-09407-f007:**
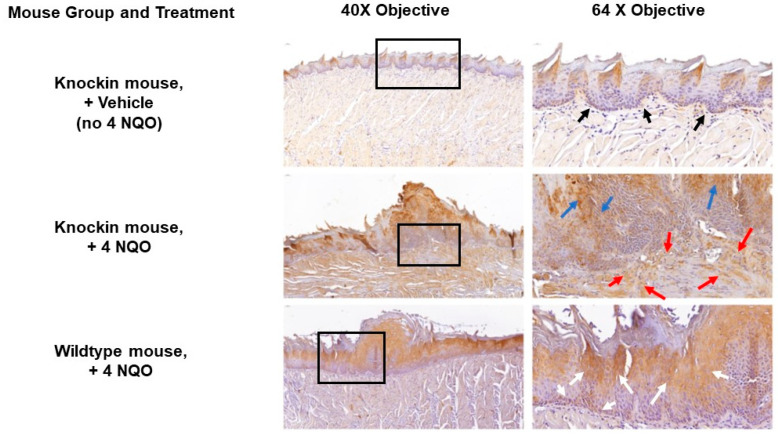
LOX expression is differentially regulated in knockin mice. Samples from knockin mice treated with vehicle show normal low expression of LOX in the basal epithelium (top panel, black arrows), and spinous epithelium. Knockin mice treated with 4 NQO exhibit strong staining in fibroblasts (middle panels, red arrows) and in the suprabasal epithelial cells (middle panel, blue arrows). Wildtype mice express LOX primarily in suprabasal cells (bottom panels, white arrows) with observable staining in basal epithelial cells. The boxes identify regions shown at higher magnification in the right column. Images are representative of 4 mice per group subjected to histology and immunohistochemistry.

**Figure 8 ijms-24-09407-f008:**
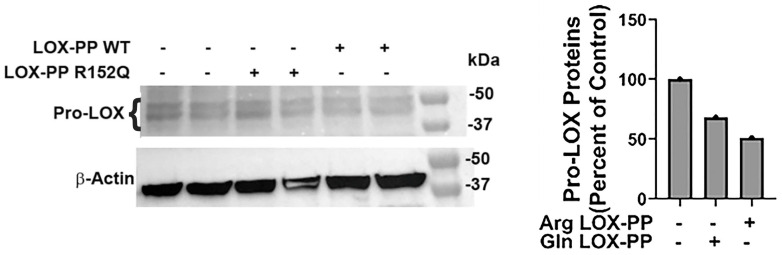
G152A polymorphism of LOX-PP attenuated the negative feedback inhibitory effect on LOX expression. Western blot of LY2 cell layers after treatment with 8 µg/mL rArg LOX-PP or rGln-LOXPP.

**Figure 9 ijms-24-09407-f009:**
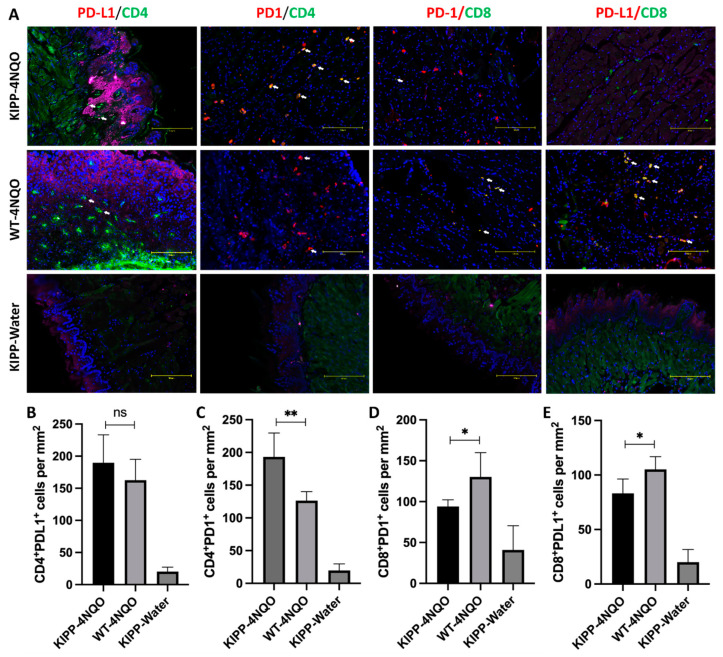
Differential expression of T cell immune checkpoint PD-L1 and PD-1 in tongue SCC lesions in wildtype (WT) Arg LOX-PP mice and knockin Gln LOX-PP (KIPP) mice. FFPE sections of 4 NQO-induced lesions or controls were stained with PD-L1, PD-1, CD4, and CD8 antibodies, and the images were captured by the EVOS M5000 system. (**A**) The representative images of CD4^+^PD-L1^+^, CD4^+^PD-1^+^, CD8^+^PD-1^+^, and CD8^+^PD-L1^+^ cells in each group. White arrows showed double-positive cells. (**B**–**E**) The numbers of double-positive cells were quantified by ImageJ and normalized in cells/mm^2^. Statistical analysis was first calculated by ANOVA and then an unpaired *t*-test was used to calculate the statistical difference between the Arg LOX-PP and Gln LOX-PP groups. The significance (*p*-value) is defined as * *p* < 0.05, ** *p* < 0.01, and ns *p* > 0.05. Scale bar, 150 µm.

**Figure 10 ijms-24-09407-f010:**
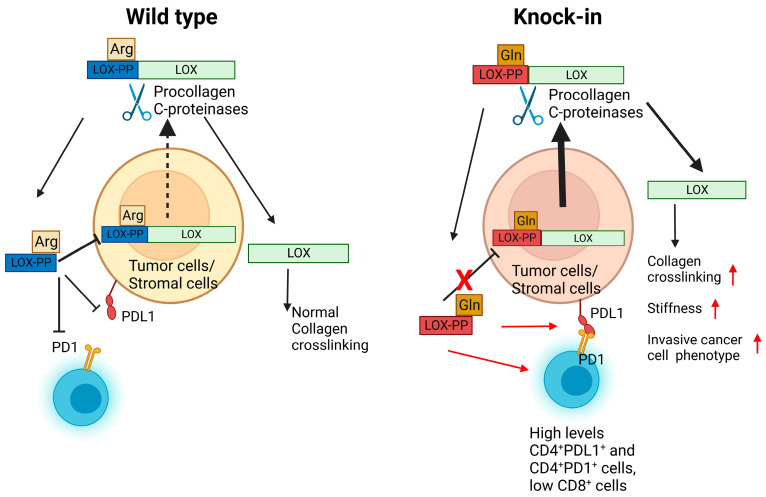
Summary and hypotheses regarding differences in activity between wildtype Arg LOX-PP and variant Gln LOX-PP in OSCC tissues. While Arg LOX-PP can inhibit LOX production through a negative feedback mechanism variant (left cartoon and Figure 8), Gln LOX-PP loses this inhibitory effect. This ultimately results in more secretion of LOX from oral lesions in ECM and progressing tumor formation (right cartoon). Additional differential effects of Gln LOX-PP occur on T cells in the microenvironment are suggested by high CD4^+^PD1^+^ and high CD4^+^PDL1^+^ T in the knockin mice (right cartoon and Figure 9), which could lead to enhanced immunosuppression. Created with BioRender.com.

**Table 1 ijms-24-09407-t001:** Tumor Incidence Among Knockin Mice and Wildtype Mice Treated with 4 NQO.

	LOX-PP Knock-in Group Treated with 4 NQO (8 Mice)	Wild Type Group Treated with 4 NQO (7 Mice)	LOX-PP Knock-in Group Treated with Water (5 Mice)
12th Week	1	0	0
16th Week	2	0	0
17th Week	3	1	0
Total	6 (75%) *	1 (14%)	0

Chi-Square test (3X2 analysis) was performed and indicated significant number of mice with oral cancer in LOX-PP knock-in group treated with 4-NQO compared to other experimental groups (*p* = 0.008). Fisher’s exact test (2X2 analysis) KIPP with 4 NQO vs. WT with 4 NQO, * *p* < 0.05 indicates that LOX-PP knock-in mice treated with 4-NQO have more incidence of cancer than wild type mice treated with 4-NQO.

## Data Availability

The data presented in this study are available on request from the corresponding author. Downloaded *LOX* sequence data are stored on the Forsyth Institute server and will be provided upon request.

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
