# Peer review of "The Lysyl Oxidase G473A Polymorphism Exacerbates Oral Cancer Development in Humans and Mice"

_ijms, 2023, doi:10.3390/ijms24119407_

Round 1

Reviewer 1 Report

The Authors present a paper entitled “The lysyl oxidase G473A polymorphism exacerbates oral cancer development in humans and mice”.  The topic is of interest to these days. The article is well written and well organized, with a clear description of the various experiments conducted both in vitro and in vivo.

.

However, some points need to be better addressed:

  • Figure 4 is confusing, the significances should be better organized, perhaps by reducing the size

of the font for all of them to have a clearer picture.

  • The initial part of the discussion, from lines 240 to 255, is an unnecessary repetition of what was written in the introduction.
  • At the end of the discussion, it would be interesting to know what the future projects are and how they plan to exploit the discoveries obtained in this work.

The text should be revised in order to avoid typos.

Author Response

Thank you for your review. 

Figure 4A has been revised to more clearly show the significance of the data points and time dependence of tongue volume measurements.

The Discussion has been revised and includes mention of future plans. Specifically, we are interested in exploring differences in immune cell profiles in the lesion microenvironments in the wildtype compared to the knockin mice. This is our near term future goal. 

Reviewer 2 Report

This article examined the role of lysyl oxidate G473A polymorphism in oral squamous cell carcinoma of human and mouse model. Several points should be considered. 

 Major points

1.       Was the pathologist joined the authors? If not, it needed. Pathological examination, one of the main stories of this article should be refined.

2.       The number of examined mouse model was relatively small (Table 1). Histological diagnosis should be made in each tumor. Papilloma/Dysplasia? Carcinoma? Invasive or in situ?

3.       It is no meaning of the experiment of tongue volume (Fig. 4). Instead, the size of tumors with histological diagnosis should be demonstrated.

4.       Figure 5 should be refined. Higher power view of histological figs needed. PCNA immunohistochemistry (IHC) could not be evaluated in this magnification.

5.       Similarly, IHC of LOXL2 and LOX should be refined with higher magnification.

6.       In Fig 5B, epithelial thickness is not essential for carcinogenesis. Histological evaluation by pathologist is needed.

7.       In Fig. 6, diminished band should be indicated by arrow head.

8.       Story of PD-1/PD-L1 was too abrupt. Detailed discussion needed.  

Minor points

      1.   P5 line 164 Figure 4 might be Figure 5. 

Author Response

  1. We have now consulted with Professor Vikki Noonan, an oral pathologist at Boston University. Consequently, we have reorganized the histology and immunohistochemistry figures, and added a more complete description of the oral lesions. The lesions in the 4 NQO treated mice are dysplastic and and not yet oral squamous cell carcinoma. This has been clarified and discussed in the revised manuscript.
  2. Data obtained in quantitative analyses were significant, and differences between features of wildtype and knockin mice after 4 NQO treatment were very robust and consistent within each group in this experimental mouse model. 
  3. The quantitative epithelium width measurements support the onset of different degrees of hyperplasia and dysplasia in the three experimental groups. Progression of  lesions through hyperplasia, followed by dysplasia and finally OSCC is a feature of development of the 4 NQO model.  At the time point chosen dysplasia is the predominant phenotype. This is more fully explained and clarified in the manuscript.  We are grateful to the reviewer for suggesting to consult with a pathologist to evaluate the histology. The manuscript is now improved. 
  4. and 5. Histology and immunohistochemistry figures have been revised and expanded and include higher magnification images and a more complete description of the histopathology.
  5. See 4 above.
  6. A pathologist was consulted, as noted above. 
  7. In the western blot figure, we have added a bracket identifying the two pro-LOX bands. Both bands are pro-LOX, the upper is glycosylated while the lower one is not. Please see JBC1992 Apr 25;267(12):8666-71.
  8. The Discussion has been modified to include the rationale for investigation of immune cells in the microenvironment.  

Round 2

Reviewer 2 Report

I agreed with the authors response to the comments.